# Determinants of incomplete childhood immunization among children aged 12–23 months in Dabat district, Northwest Ethiopia: Unmatched case- control study

**Moges Muluneh Boke**[1]*, **Getaw Tenaw**[2], **Neamin M. Berhe**[3], **Woynhareg Kassa Tiruneh**[1]

**1** Department of Reproductive Health, Institute of Public Health, College of Medicine and Health Sciences, University of Gondar, Gondar, Ethiopia, **2** Dabat Health Office, Dabat, Ethiopia, **3** Department of Public Health, ABH Campus, Jimma University, Jimma, Ethiopia

* Mogelove75@gmail.com

**Data Availability Statement:** All relevant data are within the manuscript and its Supporting information files.

## Abstract

### Background

Despite the effort to increase access to routine immunization, majority of children in low-resource countries including Ethiopia are still either unvaccinated or under-vaccinated. In Ethiopia for the past four decades, the completion rate of routine childhood immunization remains low particularly in a rural area. In this study setting, evidence regarding the socio-economic, maternal continuum care, and caregiver characteristics effect on child immunization is limited. Hence, this study aimed to identify the determinants of incomplete vaccination among children aged 12–23 months in Dabat district, Northwest Ethiopia.

### Methods

A community-based unmatched case-control study design was employed among 132 cases and 262 controls. Multi-stage sampling method was used to recruit eligible study participants. Logistic regression analysis was used to identify the determinants to children's incomplete vaccination.

### Results

Caregivers' attitude towards vaccine (AOR: 6.1, 95% CI 3.4 to 11.1), knowledge on the schedule of vaccination (AOR: 4, 95% CI 2.2 to 7.1), Place of delivery (AOR: 2.7, 95% CI 1.3 to 5.5), and marital status (AOR: 2.36, 95% CI 1.22 to 4.56) were statistically significant association with incomplete childhood vaccination.

### Conclusion

Home delivery, caregivers' poor knowledge on the schedule of vaccination, caregivers' negative perception towards vaccine and unmarried marital status were predictors to incomplete vaccination. Therefore, to enhance full vaccination coverage, immunization health education program needs to address vaccine related safety enquiries in a meaningful

**Funding:** The funders had no role in study design, data collection and analysis, decision to publish, or preparation of the manuscript.

**Competing interests:** The authors have declared that no competing interests exist.

method to caregivers, in order to improve the perception of caregivers towards vaccine. Moreover, improving maternal continuum care coverage is crucial.

## Introduction

Immunization is a crucial element of primary health care and an undisputed human right. Child immunization is one of the most successful and cost-effective public health interventions for common childhood illness like pneumonia, diphtheria, tetanus, whooping cough and measles. Nowadays, the vaccines are available to protect at least 20 diseases, and to save the life of 3 million children every year [1, 2].

The most common driving causes of under-five deaths are pneumonia, diarrhea, measles, and preterm birth complications that can be avoided or treated with simple affordable intervention such as vaccination or antibiotics [3]. Every year, globally 3.37 million children acquire pneumonia, and 140 000 measles deaths occur. The most of pneumonia cases and measles deaths occur in South Asia and Sub-Saharan Africa countries [4, 5].

Globally, an estimated 5.2 million under-five children died in 2019. The global under-five mortality rate declined by 59%, from 93 deaths per 1,000 live births in 1990 to 38 in 2019, due in huge portion of immunization [3]. However, the largest global backslide in vaccination coverage in 30 years happened recently due to the COVID-19 pandemic [3, 6, 7].

Recently, the global un or under-vaccinated children number has rapidly increased from 19.4 million in 2018 to 25 million in 2021, due to COVID-19 pandemic. More than half (60%) of these children found in ten developing countries (Ethiopia, Philippines, Democratic republic of Congo, India, Nigeria, Indonesia, Brazil, Pakistan, Myanmar, and Angola) [7].

In 1974, WHO established the Expanded Program on Immunization (EPI) to ensure every child had access to recommended vaccines. Initially, those vaccines were limited Bacillus Calmette–Guérin (BCG), diphtheria, pertussis, and tetanus (DPT), and oral poliovirus. EPI started in Ethiopia in 1980 to reducing morbidity and mortality of children from vaccine-preventable diseases. According to the national Immunization Implementation Guideline, children are considered completely vaccinated when they are provided a vaccination against tuberculosis (BCG) at birth, three doses of DPT-HepB-Hib at 6, 10 and 14 weeks, four doses of polio vaccines at birth, 6, 10 and 14 weeks, three doses of PCV at 6, 10 and 14 weeks, two doses of rotavirus vaccine at 6, 10 weeks and a measles vaccination by the age of 9 months [8].

In last decade, Ethiopia government implemented four main strategic areas and the monitoring and evaluation, accountability frame work of Global Vaccine Action Plan (GVAP). The framework included services: vaccine supply, service delivery, quality and logistics, disease surveillance and control, advocacy, and social mobilization to attain at least 90% national coverage and Sustainable Development Goals (SDGs) target "leave non one behind". However, full completion rate of routine childhood immunization remains low [8, 9].

In Ethiopia, according to current systematic review and meta-analysis report, one child in two children not vaccinated or four out of ten children have incomplete vaccine. The magnitude of incomplete vaccination substantially varied across the subnational and local levels. In Northwest Ethiopia, it ranges from 7–24%. In this study setting, 23.1% children had incomplete vaccination [10]. In existing literatures, associated factors maternal illiteracy, fear of side effects, rural residence and home delivery were increased incomplete vaccination [11, 12]. However, factors like wealth status, maternal autonomy, and attending maternal health care services were reduced incomplete vaccination [10, 13–15]. Other contextual factors, such as

sociocultural beliefs, perception for vaccine influence the behavior of caregivers, and affect the completion of vaccination. However, the effect of sociocultural beliefs, and caregivers' perception influence on vaccine completion were not well known. In order to improve the coverage of vaccine and to develop tailored intervention strategy, identifying and understanding barriers of caregivers to accessing immunization services is essential. Therefore, this study aimed to assess determinants of incomplete immunization among children aged 12–23 months in Dabat district, northwest Ethiopia.

## Methods and materials

### Study design, and setting

A Community based unmatched case control study design was conducted with ratio of case to control 1:2 in Dabat district, from April to May 2021. Dabat district found in Northwest Ethiopia. It has 4 urban, and 32 rural kebeles (the smallest unit of administration); 145,458 total population, and 17,112 children age 12–23 months. The district has six health centers, one primary hospital and 35 health posts and more than 650 health care providers.

### Source and study population

All children aged 12–23 months with their caretakers and who had taken at least one dose of routine vaccination were considered as a source population. Study populations were children aged 12–23 months with mothers/caregivers residing in randomly selected kebele's of the district in northwest Ethiopia. Cases were children aged 12–23 months and defaulted recommended vaccination before their 1st birthday. In contrast, those children aged 12–23 months and completed the recommended vaccination before their 1st birthday were considered as controls. Caregivers who were critically ill, and unable to communicate were excluded from the study.

### Sample size determination, and sampling procedure

To estimate the sample size; 95% confidence level, 80% power, 2 design effect, case to control ratio of 1:2 and 10% of non-response rate assumptions were used. Moreover, to estimate the sample size all significantly associated factors in previous studies were considered [12, 15–20]. Finally, using independent variable not attending postnatal care (PNC), 394 (132 cases and 262 controls) sample size was estimated.

Multi-stage sampling technique was employed to select cases and controls. Initially, the kebeles in the district were stratified into urban and rural. Six rural and two urban kebeles were randomly selected by a lottery method from each stratum. To identify all eligible cases and controls health facility the EPI registration books were used. Then, proportional allocation of simple size employed to each selected kebeles based on number of eligible children. Finally, to get study participants from selected kebele simple random sampling method were employed.

### Operational definition of variables

**Complete vaccination.** A child who received ten basic vaccines (one dose of BCG, three doses each of the DPT-HepB-Hib (pentavalent), three doses of polio vaccines, three doses of PCV, two doses of Rota vaccine, and one dose of measles vaccine before her/his 1st birthday) was considered as completely vaccinated [21].

**Incomplete vaccination.** A child between 12–23 months old who missed at least one dose of the ten vaccines before their 1st birthday [17].

**Knowledge at schedule of immunization.** To assess the knowledge of caregivers were used four items' questions. Caregivers who score greater than 50% were categorized as good knowledge [22].

**Attitude towards benefit of vaccination.** Six perception related questions were asked using Likert scale items. Each item has five response options. Caregivers who score 70% and above were categorized as positive attitude [22].

**Misconceptions about vaccine contraindication.** Six misconception related questions were asked using Likert scale tool. Each item has five response options. Caregivers who score 70% and above were categorized as having misconception [22].

## Data collection instrument and procedure

To collect data, pretested, structured interviewer administered questionnaires were used. A questionnaire was developed through reviewing different literatures [10–13, 15–19, 21–23]. The questionnaire initially developed in English and translated to local Amharic language, and then translated back to English language to check consistency.

The questionnaire has three sections; socio-demographic characteristics, obstetric history, and health service-related variables. Six Bsc nurses and two public health officers were involved on data collection and supervision activities. Caregivers who were not present at home during the first day of data collection were revisited in the subsequent days until the final day of data collection.

To ensure quality of data, training was given for data collectors and supervisor on data collection technique. The Pretest was done on 5% (20) of sample size at Wegera district and some minor modifications were made after pretest. The data collection process was supervised by supervisor and principal investigator daily. At the end of each data collection day, the principal investigator and supervisors checked the completeness of questionnaires daily.

## Statistical analysis

Data were checked for completeness and consistency before entered into EpiData version 4.6 software and then exported to SPSS version 20 for further analysis. Data were cleaned for missing value, outlier, and inconsistency before analysis. Descriptive statistics such as means, median, proportion, percentage, and interquartile range were computed and presented in text, table and graph. To assess the association of covariates and dependent variables bi-variable logistic regression analysis was performed. A p-value of less than 0.05 and Adjusted Odds Ratios (AOR) with a 95% confidence interval (CI) were used to report significantly associated variables. Moreover, Hosmer and Lemeshow goodness fitted to check model fitness and variance inflation factor (VIF) to assess multi-collinearity were also performed. To estimate household wealth status principal component analysis was done. Finally, the wealth status was ranked into three quantiles.

## Ethics approval and consent to participants

Ethical clearance was obtained from ethics committee of the University of Gondar, College of Medicine and Health Science, Institute of Public Health. Permission letter to conduct the study was also be taken from Dabat Woreda Health office. Before data collection, adequate information was given on study procedure, data storage, benefit, privacy concern and voluntary participation. Verbal informed consent was obtained from study participants before data collection. In the Informed consent the purpose of study, risk and benefit of the study, procedures, interview taking time, and the right of participants rights were

explained. To ensure study participants confidentiality and privacy, the data was stored in separate computer and the access was restricted using password. Furthermore, the personal name and other identification of the participants were not recorded on the data collection format.

## Results

### Socio-demographic and economic characteristics of caregivers

From a total of 394 children, 383 children's caregivers (127 cases and 256 controls) were involved, with a response rate of 96.2% of cases and 97.7% controls. The mean age of caregivers for cases and controls were 31.37±8.27 and 30.66±8.4 years respectively. The majority of caregivers for cases 119(93.7%) and controls 243 (94.9%) were mothers. Regarding marital status, about 41 (32.3%) cases and 50 (19.5%) controls were unmarried. Most case 88 (69.3%) and controls 168(65.6%) were from rural communities. The educational status of caregivers; 74 (58.3%) of cases and 134 (52.3%) of controls were no formal education (Table 1).

### Obstetrics history

Regarding birth interval the median birth interval was 24 months for cases and 25 months for controls. Birth order 43 (35.9%) of cases and 41 (16%) of controls were in the birth order of fourth and above. The majority of mothers, 56.7% of cases and 37.5% of controls had three or more parity (Table 2).

### Maternal continuum care service-related characteristics

Regarding antenatal care follow-up, 53 (41.7%) of cases and 162 (63.3%) of controls attended at least one ANC follow-up during their index pregnancy. Place of delivery, 45 (35.4%) of cases and 182 (71.1%) of controls give birth at health institutions. Less than half of cases (39.4%), and 56.6% controls were received at least one post-natal care. From post-natal attended women more half of women were not received health education (Table 3).

### Caregiver knowledge, attitude, and perception of immunization

The majority of incomplete immunized children's caregivers (65.4%) had a poor knowledge on the schedule of vaccination, and negative attitudes to vaccines benefits (63.8%). The misconception of caregiver to immunization is the reason to not vaccinate the child. In this study, 48% of cases and 46.1% of controls had the misconception on vaccine benefits. Travelling longer distance to the vaccination site influences the vaccine uptake, in this study setting nearly half of cases (48.8%), and controls (46.9%) travel more than 30 minutes to reach the vaccination site, Table 4. The main source of vaccine information for cases was television (40.3%), for controls was health care provider (35.6%) (Fig 1).

### Caregivers' determinants to childhood incomplete vaccination

In binary logistic regression analysis; covariates marital status, place of delivery, knowledge on schedule of vaccination and perception towards benefits of vaccine were significantly associated with incomplete immunization of a child.

The odds of incomplete child immunization were 2.36 times higher among unmarried women as compared to married women (AOR: 2.36, 95% CI: (1.22, 4.56)). Children who were born at home had 2.7 times more likely incomplete vaccinated compared to health institutions (AOR: 2.7, 95% CI: (1.3, 5.5)). The odds of incomplete child immunization were 4 times higher

**Table 1. Socio-demographic characteristics of caregivers in Dabat district, Northwest, Ethiopia, 2021.**

| Variables | Cases, n (%) | Controls, n (%) | P-value |
|---|---|---|---|
| **Place of residence** | | | |
| Rural | 88 (69.3) | 168 (65.6) | 0.491 |
| Urban | 39 (30.7) | 88 (34.4) | |
| **Caregivers** | | | |
| Mother | 119 (93.7) | 243 (94.9) | 0.541 |
| Others | 8 (6.3) | 13 (5.1) | |
| **Caregivers age** | | | |
| < = 25 years | 29 (22.8) | 82 (32) | 0.97 |
| 26–34 years | 62 (48.8) | 102 (39.8) | |
| > = 35 years | 36 (28.3) | 72 (28.1) | |
| **Marital status** | | | |
| Unmarried | 41 (32.3) | 50 (19.5) | 0.041 |
| Married | 86 (67.7) | 206 (80.5) | |
| **Caregivers' educational status** | | | |
| No formal education | 74 (58.3) | 134 (52.3) | 0.025 |
| Primary school | 37 (29.1) | 66 (25.8) | |
| Secondary and above | 16 (12.6) | 56 (21.9) | |
| **Caregivers' occupation** | | | |
| Farmer | 26 (20.5) | 51 (19.9) | 0.052 |
| Merchant | 27 (21.3) | 34 (13.3) | |
| Daily laborer | 11 (8.7) | 15 (5.9) | |
| House wife | 51 (40.2) | 134 (52.3) | |
| Gov't employer | 12 (9.4) | 22 (8.6) | |
| **Husband educational status** | | | |
| No formal education | 45 (52.3) | 123 (59.7) | 0.075 |
| Primary school | 20 (23.3) | 35 (17) | |
| Secondary school and above | 21 (24.4) | 48 (23.3) | |
| **Child sex** | | | |
| Female | 64 (50.4) | 132 (51.6) | 0.829 |
| Male | 63 (49.6) | 124 (48.4) | |
| **Age of the child (month)** | | | |
| 12–18 | 89 (70.1) | 171 (66.8) | 0.035 |
| 19–23 | 38 (29.9) | 85 (33.2) | |
| **Family size in the household** | | | |
| > = 5 | 70 (55.1) | 81 (31.6) | 0.000 |
| < 5 | 57 (44.9) | 175 (68.4) | |
| **Wealth index** | | | |
| Poor | 34 (26.8) | 90 (35.2) | 0.025 |
| Middle | 46 (36.2) | 60 (23.4) | |
| Rich | 47 (37) | 106 (41.4) | |

Others = Grand-parents, Fathers and Siblings.

among caregivers who had poor knowledge on the schedule of vaccination (AOR: 4, 95% CI: (2.2, 7.1)). The odds of incomplete child immunization were 6.1 times higher among caregivers who had unfavorable attitude towards the benefits of the vaccine as compared to counter (AOR: 6.1, 95% CI: (3.4, 11.1)) (Table 5).

**Table 2. Obstetrics characteristics of mothers in Dabat district, Northwest, Ethiopia, 2021.**

| Variables | Cases, n (%) | Controls, n (%) | P-value |
|---|---|---|---|
| **Parity** | | | |
| One | 24 (18.9) | 93 (36.3) | 0.000 |
| Two | 31 (24.4) | 67 (26.2) | |
| Three and above | 72 (56.7) | 96 (37.5) | |
| **Birth order** | | | |
| 1 | 24 (18.9) | 93 (36.3) | 0.036 |
| 2–3 | 60 (47.2) | 122 (47.7) | |
| > = 4 | 43 (33.9) | 41 (16) | |
| **Birth interval in months** | | | |
| < 24 | 45 (43.7) | 59 (36.2) | 0.723 |
| 24–36 | 49 (47.6) | 92 (56.4) | |
| > = 37 | 9 (8.7) | 12 (7.4) | |
| **Types of index pregnancy** | | | |
| Wanted | 76 (59.8) | 163 (63.7) | 0.466 |
| Unwanted | 51 (40.2) | 93 (36.3) | |

## Discussion

Childhood mortality can be significantly lowered by providing essential vaccine to the child. However, enormous number of eligible children in Ethiopia are un vaccinated or under vaccinated due to contextual factors and other barriers. Moreover, currently the COVID 19 pandemics backslide in vaccination coverage in the world [3, 5, 7, 9].

This study revealed several determinants to incomplete childhood immunization. Marital status being unmarried, home place of delivery, poor knowledge of caregivers on vaccine

**Table 3. Maternal continuum care characteristics mothers in Dabat district, Northwest, Ethiopia, 2021.**

| Variables | Cases, n (%) | Controls, n (%) | P-value |
|---|---|---|---|
| **Visited by health worker during post-partum period** | | | |
| Yes | 50 (39.4) | 145 (56.6) | 0.001 |
| No | 77 (60.6) | 111 (43.4) | |
| **Tetanus toxoid vaccination during pregnancy** | | | |
| Yes | 55 (43.3) | 155 (60.5) | 0.001 |
| No | 72 (56.7) | 101 (39.5) | |
| **Antenatal care** | | | |
| Yes | 53 (41.7) | 162 (63.3) | 0.000 |
| No | 74 (58.3) | 94 (36.7) | |
| **Postnatal care** | | | |
| Yes | 52 (40.9) | 179 (69.9) | 0.000 |
| No | 75 (59.1) | 77 (30.1) | |
| **Attended health education during the post-partum period** | | | |
| Yes | 55 (43.3) | 127 (49.6) | 0.245 |
| No | 72 (56.7) | 129 (50.4) | |
| **Place of delivery** | | | |
| Home | 82 (64.6) | 74 (28.9) | 0.000 |
| Health institution | 45 (35.4) | 182 (71.1) | |

**Table 4. Caregiver knowledge, attitude, and perception of immunization in Dabat district, Northwest, Ethiopia, 2021.**

| Variables | Cases, N (%) | Controls, N (%) | P-value |
|---|---|---|---|
| **Time is taken to vaccination site (in a minute)** | | | |
| < 30 | 65 (51.2) | 136 (53.1) | |
| > = 30 | 62 (48.8) | 120 (46.9) | |
| **Child immunization card** | | | |
| Yes | 69 (54.3) | 164 (64.1) | 0.720 |
| No | 58 (45.7) | 92 (35.9) | |
| **Mothers' role in the community** | | | |
| Leaders of HDA | 30 (23.6) | 66 (25.8) | 0.704 |
| Members of HDA | 53 (41.7) | 112 (43.8) | |
| Not members of HDA | 44 (34.6) | 78 (30.5) | |
| **Knowledge on schedule of vaccination** | | | |
| Good | 44 (34.6) | 174 (68) | 0.00 |
| Poor | 83 (65.4) | 82 (32) | |
| **Misconception about vaccine** | | | |
| Yes | 61 (48) | 118 (46.1) | 0.720 |
| No | 66 (52) | 138 (53.9) | |
| **Perception towards side effect of vaccine** | | | |
| Positive | 46 (36.2) | 190 (74.2) | 0.000 |
| Negative | 81 (63.8) | 66 (25.8) | |

Note, "HAD" = Health development army

schedule and unfavorable attitude to vaccine were determinant to incomplete immunization of child.

In the current study the single marital status of mothers increased the risk of under vaccination of child compared to married mothers. This finding is supported by studies conducted in Minjar [24], Togo [25], and birth cohort study of Japan [23]. The possible reason for this might be social background of married and unmarried mothers. Married women have strong social bondage than unmarried mothers, and more likely discuss about child vaccination with their spouse and peers, which might be help mothers to remember vaccine schedule [24]. This evidence also supported by study conducted Alberta, Canada. Mothers who had timely and completely vaccinated their infants were positively influenced by experiences of their family or peers [26].

From component of continuum maternal care, not giving birth in the health institution were increased the risk of incomplete immunization. Children who were born at home more likely incomplete vaccinated. This finding agrees with the studies conducted in East Gojjam [18], Nigeria [25] and Northwest Ethiopia [12]. In fact, during institution delivery women are sensitized on the child health care including vaccination benefits and schedules and newborn receive polio zero and BCG vaccine. However, in Ethiopia the coverage of health institution delivery is low [27]. Hence, the finding of this study suggest that Ministry of Health need to focus on health institutional delivery to improve the uptake childhood immunization, maternal and child health outcomes.

Lack of knowledge on the schedule of vaccination increased the risk of childhood immunization defaulters. Caregivers who had poor knowledge on schedule of vaccination more likely defaulted child immunization than counterpart. The finding of this study confirmed studies

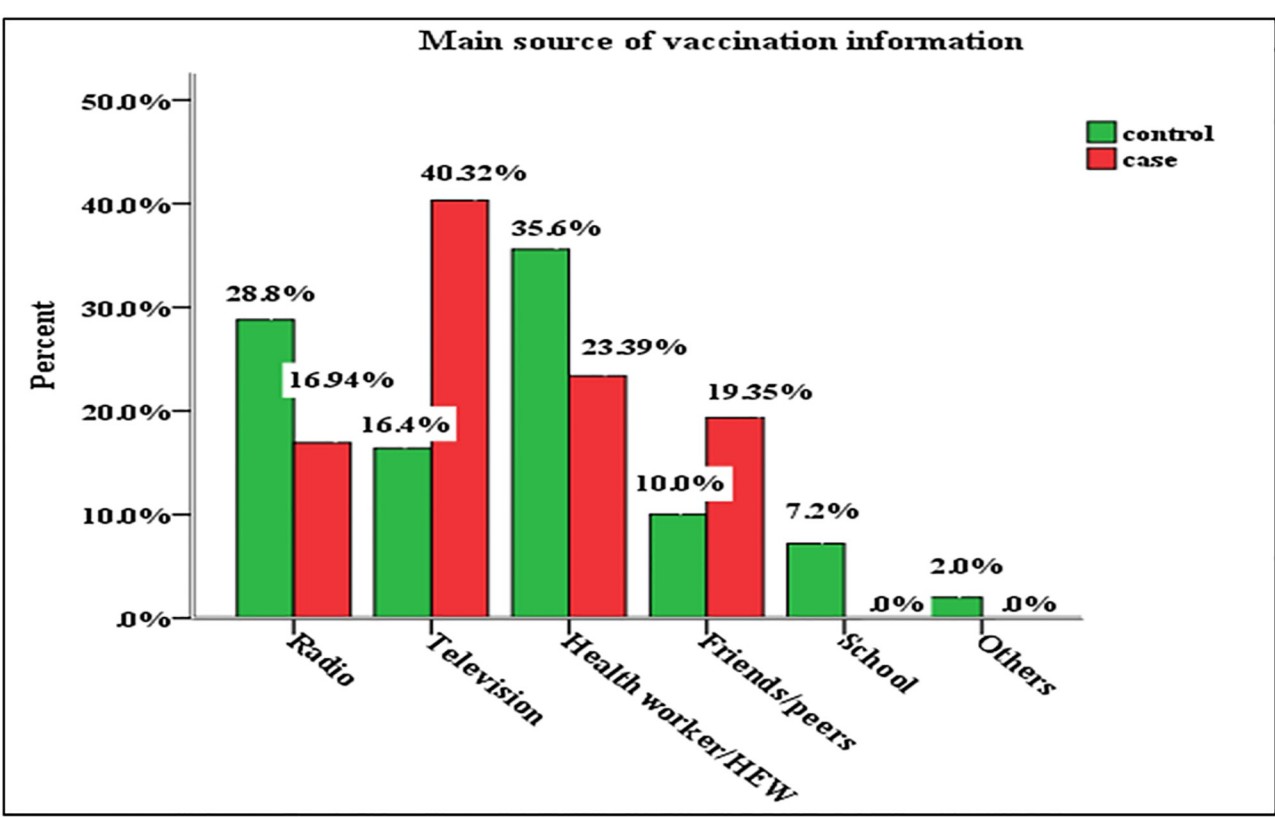

**Fig 1. Caregivers source of information for vaccination in Dabat district, Northwest, Ethiopia, 2021.**

result obtained from Gondar, and Hawassa zuria district [12, 17]. In fact, better knowledge about vaccine schedule and benefits will motivate to immunize the child.

Regarding contextual factors, such as sociocultural beliefs and caregivers' attitude to vaccine. The odds of incomplete child immunization higher among caregivers who had unfavorable attitude towards vaccine compared to counterpart. This finding in line with the studies result obtained from East central Ethiopia [28], Cameroon [29], and Somalia [30]. This might be mother who had negative perception on vaccine, less confident on vaccine safety and wrongly outweigh the risks benefits of vaccine, as result prefer not vaccinate [26, 31]. This evidence is also supported by study finding obtained from Gondar, caregiver who had fear of vaccine side effect more likely default child immunization as compared to no fear [13]. Therefore, immunization health education needs to address vaccine related safety enquiries in meaningful method to caregivers, in order to improve the perception of caregiver on vaccine as well as the coverage of vaccination.

The strength and limitation of the study, in this study the perception of the caregiver toward vaccine was assessed, which critical for designing other intervention approaches and to decrease immunization defaulter rate. However, the perception of caregivers to vaccine were not explored in-depth to understanding to contextual related barriers of caregivers and heath institutions using qualitative study. Moreover, this study was used immunization card, and EPI records to minimize recall bias and misclassification of age and immunization status of the child. However, data related with continuum maternal care such as antenatal and postnatal care number of visits for index pregnancy, were collected by self-report which might be prone to recall and socially desirable bias.

**Table 5. Determinants of incomplete immunization among children 12–23 months in Dabat district, Northwest, Ethiopia, 2021.**

| Variables | COR (95%CI) | AOR (95%CI) |
|---|---|---|
| **Marital status** | | |
| Unmarried | 1.96(1.21–3.18) | **2.36(1.22–4.56)**\* |
| Married | 1 | 1 |
| **Caregivers' educational status** | | |
| No formal education | 1.93(1.03–3.6) | 0.51(0.19–1.34) |
| Primary school | 1.96(0.98–3.89) | 0.76(0.3–1.95) |
| Secondary and above | 1 | 1 |
| **Family size** | | |
| > = 5 | 2.65(1.71–4.11) | 2.08(0.8–5.36) |
| < 5 | 1 | 1 |
| **Wealth index** | | |
| Poor | 0.85(0.5–1.43) | 0.54(0.27–1.07) |
| Middle | 1.72(1.03–2.89) | 1.67(0.85–3.29) |
| Rich | 1 | 1 |
| **Number of parities** | | |
| 1 | 0.34(0.2–0.59) | 0.33(0.09–1.09) |
| 2 | 0.61(0.36–1.04) | 1.19(0.45–3.12) |
| > = 3 | 1 | 1 |
| **Birth order** | | |
| 1 | 1 | 1 |
| 2–3 | 1.9(1.1–3.28) | 0.74(0.31–1.75) |
| > = 4 | 4.06(2.18–7.55) | 1.3(0.42–1.5) |
| **Visited by health worker during post-partum period** | | |
| Yes | 1 | 1 |
| No | 2.01(1.3–3.1) | 1.27(0.71–2.2) |
| **Tetanus toxoid vaccination during pregnancy** | | |
| Yes | 1 | 1 |
| No | 2(1.3–3.09) | 0.4(0.07–2.18) |
| **Antenatal care** | | |
| Yes | 1 | 1 |
| No | 2.4(1.5–3.7) | 4(0.71–22.4) |
| **Post-natal care** | | |
| Yes | 1 | 1 |
| No | 3.35(2.1–5.2) | 1.6(0.79–3.2) |
| **Place of delivery** | | |
| Home | 4.48(2.84–7.05) | **2.7(1.3–5.5)**\* |
| Health institution | 1 | 1 |
| **Child immunization card** | | |
| Yes | 1 | 1 |
| No | 1.49(0.97–2.3) | 1.7(0.99–3.09) |
| **Knowledge on schedule of vaccination** | | |
| Good | 1 | **1** |
| Poor | 4(2.55–6.27) | 4(2.2–7.1)\* |
| **Perception towards vaccine side effect** | | |
| Positive | 1 | 1 |
| Negative | 5.06(3.2–8.01) | 6.1(3.4–11.1)\* |

\* = P-value < 0.05, 1 = Reference group

## Conclusion

In overall, this study sought health service, and contextual related determinants to childhood vaccination dropout. Furthermore, marital status being unmarried, home place of delivery, poor knowledge of caregivers on vaccine schedule and unfavorable perception to vaccine were determinant to default vaccination among the of child. Therefore, to enhance the fully immunization coverage; improving maternal continuum care coverage, and immunization health education program needs to address vaccine related safety enquiries in meaningful method to caregivers, in order to improve the perception of caregiver on vaccine.

## Supporting information

**S1 File. The datasets of the study.**
(SAV)

## Acknowledgments

First, we would like to thank University of Gondar, College of Medicine and Health Sciences, Institute of Public Health, department of reproductive health to giving me this chance. Secondly, we would like also to express our appreciation to heath department of Dabat, data collectors, supervisors and participants.

## Author Contributions

**Conceptualization:** Moges Muluneh Boke, Getaw Tenaw.

**Formal analysis:** Moges Muluneh Boke, Neamin M. Berhe, Woynhareg Kassa Tiruneh.

**Investigation:** Getaw Tenaw, Neamin M. Berhe.

**Methodology:** Moges Muluneh Boke, Neamin M. Berhe, Woynhareg Kassa Tiruneh.

**Project administration:** Getaw Tenaw.

**Software:** Moges Muluneh Boke.

**Supervision:** Getaw Tenaw.

**Validation:** Neamin M. Berhe.

**Visualization:** Getaw Tenaw.

**Writing – original draft:** Moges Muluneh Boke, Getaw Tenaw, Neamin M. Berhe.

**Writing – review & editing:** Moges Muluneh Boke, Neamin M. Berhe, Woynhareg Kassa Tiruneh.

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
