## [Decision Letter · Decision Letter 0]

28 Apr 2022

PONE-D-21-34551

Determinants of incomplete childhood immunization among children aged 12-23 months in dabat district, Northwest Ethiopia: Unmatched Case- control Study

PLOS ONE

Dear Dr. Tiruneh,

Thank you for submitting your manuscript to PLOS ONE. After careful consideration, we feel that it has merit but does not fully meet PLOS ONE’s publication criteria as it currently stands. Therefore, we invite you to submit a revised version of the manuscript that addresses the points raised during the review process.

Please look at  the suggestions  and recommendations and respond them, doing the changes or explainning why not. An extensive Englisg revision must be done by a native speaking person or  a certified translater.

We look forward to receiving your revised manuscript.

Kind regards,

Ricardo Q. Gurgel, PhD

Academic Editor

PLOS ONE

A clean copy of the edited manuscript (uploaded as the new *manuscript* file).

4. PLOS requires an ORCID iD for the corresponding author in Editorial Manager on papers submitted after December 6th, 2016. Please ensure that you have an ORCID iD and that it is validated in Editorial Manager. To do this, go to ‘Update my Information’ (in the upper left-hand corner of the main menu), and click on the Fetch/Validate link next to the ORCID field. This will take you to the ORCID site and allow you to create a new iD or authenticate a pre-existing iD in Editorial Manager. Please see the following video for instructions on linking an ORCID iD to your Editorial Manager account: https://www.youtube.com/watch?v=_xcclfuvtxQ.

5. Thank you for submitting the above manuscript to PLOS ONE. During our internal evaluation of the manuscript, we found significant text overlap between your submission and the following previously published works, some of which you are an author.

- https://www.tandfonline.com/doi/full/10.1080/21645515.2018.1502528

- https://journals.plos.org/plosone/article?id=10.1371/journal.pone.0246018

- https://pubmed.ncbi.nlm.nih.gov/33243208/

- https://media.neliti.com/media/publications/7191-EN-predictors-of-immunization-defaulting-among-children-age-12-23-months-in-hawassa.pdf

- https://www.tandfonline.com/doi/abs/10.1080/21645515.2019.1670124?journalCode=khvi20

Please revise the manuscript to rephrase the duplicated text, cite your sources, and provide details as to how the current manuscript advances on previous work. Please note that further consideration is dependent on the submission of a manuscript that addresses these concerns about the overlap in text with published work.

Reviewers' comments:

Reviewer's Responses to Questions

**Comments to the Author**

1. Is the manuscript technically sound, and do the data support the conclusions?

Reviewer #1: Yes

Reviewer #2: Yes

2. Has the statistical analysis been performed appropriately and rigorously? 

Reviewer #1: Yes

Reviewer #2: Yes

3. Have the authors made all data underlying the findings in their manuscript fully available?

Reviewer #1: Yes

Reviewer #2: Yes

4. Is the manuscript presented in an intelligible fashion and written in standard English?

Reviewer #1: Yes

Reviewer #2: No

5. Review Comments to the Author

Reviewer #1: Most of the reasons for the incomplete immunization are same as in most of the studies. No doubt it confirms that the district in study of Ethiopia is also having similar reasons. As I understand poverty and illiteracy are the two main reasons in all the third world countries.

Reviewer #2: MAJOR REVISION

The authors present a study into factors explaining incomplete vaccination in the Dabat region of Ethiopia. This is an important topic. I have several questions regarding some metholological choices made by the authors, and suggestions to improve grammar and spelling. Overall, however, it is a good study that should be part of the global evidence base regarding demand-side issues of incomplete vaccination.

Major points

- There are grammatical errors throughout the manuscript. I've tried to flag them but it may be smart to enlist the help of a text editor or native english speaker.

- What about missed opportunities of vaccination, i.e. children who did not get their vaccination at the exact right time for example due to a missed appointment, but a little while later?

- The authors state that "All eligible Cases and controls were identified by using health post EPI registration books of the selected rural kebeles and health center EPI registration books of the selected urban kebeles". This means children that are not EPI registration books cannot be part of the study as cases or controls. Can this have led to selection bias? What percentage of children born in the region are in the EPI books? If many are not, then the authors may be missing factor for incomplete immunization that are relevant for those children.

- This study focused mainly on factors relating to the mother. In earlier reports (from Nigeria), the husbands role as well as their 'permission' to give vaccination was also important: https://onlinelibrary.wiley.com/doi/pdfdirect/10.1111/j.1440-1754.2010.01956.x. Why did the authors focus mainly on the mothers? Does caregiver, for example also imply fathers, or is the impact of fathers likely to be different in Ethiopia compared to Nigeria?

- Why was the Dabat region chosen specifically for this study?

- in tables 1-4 add p values for statistical tests checking if groups differed significantly on characteristics. For example caregiver age and wealth index seems to differ, would be good to know if these are significant differences. I think standard practice is to use t tests for continuous variables and chi-square tests for categorical ones.

- at various points the age of child is mentioned. When was this measured? I assume it is the child's age at middlepoint of your study time (april/may 2021), so 1 may 2021?

- Many continous independent variables were dichotomized, what was the reasoning behind this and how are the results if this is not done?

- were there any missing data, or cases were respondents did not recall particular things? it is mentioned as a limitation but the extent to which it occured cannot be seen in the results.

Minor points

- ll 26: revise grammar of sentence

- ll 29: change to "a community-based (...)"

- ll 32 "one dose routine" -> "one dose of the routine"

- abstract results: I prefer the notation (AOR: 2.3, 95% CI 1.3 to 5.5) for clarity but refer to journal guidelines to see what they prefer

- abstract results: perhaps present these in order of largest effect to smallest effect?

- ll 41 "children's whose place of delivery at home" -> "children who were delivered at home"

- ll 49: an->the

- ll 49: "proposed to provoke" -> "which provokes"

- ll 54: "cost effective" -> cost-effective

- ll 55 please provide a reference

- ll 58 children->child

- ll 59 please specify the abbreviations

- ll 61 "vaccine preventable" -> vaccine-preventable

- ll 61 "according to national" -> "according to the national"

- ll 61: Why does EPI reduce mortality and morbidity of the 'mother' from vaccinate preventable diseases? Later section the authors show TT (tetanus injection). Maybe the can explain that here

- ll 62 remove "as"

- ll 68-80 I would suggest to move a large part of this to the start of the introduction.

- ll 70: please provide a reference for "due in large portion to immunization"

- ll 81 please specify abbreviation

- ll 86 please specify that you are talking about Ethiopia

- ll 93 "reasons of not to" -> "reasons to not"

- ll 102 perhaps it is more insightful to mention total number of births in this time period?

- ll 100-102 can you explain the different between "kebels" and "kebeles" for those unaware (like myself) what they mean?

- ll 105: consider splitting this sentence up as it is very long and therefore hard to read. For example "E.g. All children aged between 12-23 months with mothers/ caretakers who had at least one dose of routine vaccination residing in Dabat district, were considered as a source population for both cases and controls. Those children aged between 12-23 months who did not complete the recommended vaccination before her/his 1ste birthday in selected kebels during the data collection were considered cases. Whereas those children aged between 12-23 months who completed the recommended vaccination before her/his 1ste birthday in selected kebeld during data collection period were considered as controls"

- ll 108 "her/his" -> "their"

- ll 110 "her/his" -> "their"

- ll 113 "from EPI" -> "from the EPI"

- ll 118 "for unmatched" -> "for an unmatched"

- ll 121 "a lot of" -> "many"

- ll 121 specify abbreviation

- ll 121 "follows up" -> "follow-ups"

- ll 122 remove "so,"

- ll 127 "by lottery" -> "by a lottery"

- ll 134 "Socio" -> "socio" (remove capital)

- ll 134: I take it these are independent variables as well? This is not immediately clear from the text.

- ll 138-139 please specify abbreviations

- ll 133-169 this may be converted to tables for clarity

- ll 149 "his/her" -> "their"

- ll 151: what about delayed vaccination i.e. missed opportunities?

- ll 171-172 remove "which were published in credential journals"

- ll 173 "in to" -> "into"

- ll 189 remove "were"

- ll 212 level->levels

- table 1: at occupation for controls, the 98.6% at gov't seems like an error as the percentages add up to more than 100%

- ll 283 remove "were"

- ll 240 Sorry I do not understand, round of what?

- ll 241 remove "to"

- table 3 please specify abbreviations.

- ll 248 "had good knowledge" -> "had a good knowledge"

- ll 253 I think this belongs in the caption of table 4?

- table 5 please specify abbreviations seperately in the caption

- table 5 first two columns (cases and controls) can be removed here as they already reported in earlier tables.

- ll 273 "but" -> "but a"

- ll 274 remove "was"

- ll 275 "increase them to die" -> "increases their risk of dying"

- ll 284 "also help by remembers" -> "can also help by remembering"

- ll 286 "was significant" -> "was a significant"

- ll 294 "was significant" -> "was a significant"

- ll 301 which->this

6. PLOS authors have the option to publish the peer review history of their article (what does this mean?). If published, this will include your full peer review and any attached files.

Reviewer #1: **Yes: **Dr Raju Shah

Reviewer #2: **Yes: **Henk Broekhuizen

---

## [Author Response · Author response to Decision Letter 0]

12 Aug 2022

To: Editor-in-chief of PLOS ONE Journal

Subject: Submission of a revised manuscript for publication

Dear Editor,

Thank you for allowing us to submit a revised draft of our manuscript entitled “Determinants of incomplete childhood immunization among children aged 12-23 months in Dabat district, Northwest Ethiopia: Unmatched Case- control Study" [Manuscript ID number: PONE-D-21-34551]. We appreciate the time and effort dedicated by you and the reviewers to provide your valuable feedback on our manuscript. We are grateful to the reviewer for their insightful comments which improve our manuscript. We accepted and tried to incorporate all of the comments provided. Thus, the comments are attached here below with their point-by-point responses given in blue font color. Besides, thedetailed changes made are highlighted in the “revised manuscript with track changes” to easily identify the changes/improvements, and clean copy of the revised manuscript are prepared.

Response for Academic editor 

1. We suggest you thoroughly copyedit your manuscript for language usage, spelling, and grammar. If you do not know anyone who can help you do this, you may wish to consider employing a professional scientific editing service.

Response: Thank you for your comments, we have corrected the grammar, and spelling errors. The language edited by Berihun Melaku. He is Assistant professor of Epidemiology. 

Response: Thank dear editor for your comments. We want to update our data availability statement. We have also included amendment statement on the cover letter.

3. Thank you for submitting the above manuscript to PLOS ONE. During our internal evaluation of the manuscript, we found significant text overlap between your submission and the following previously published works, some of which you are an author.

https://www.tandfonline.com/doi/full/10.1080/21645515.2018.1502528
https://journals.plos.org/plosone/article?id=10.1371/journal.pone.0246018
https://pubmed.ncbi.nlm.nih.gov/33243208/
https://media.neliti.com/media/publications/7191-EN-predictors-of-immunization-defaulting-among-children-age-12-23-months-in-hawassa.pdf

https://www.tandfonline.com/doi/abs/10.1080/21645515.2019.1670124?journalCode=khvi20

Response: Thank dear editor for you comments. We have paraphrase and rewrite the all paragraphs and section of the manuscript. Please see the revised version of manuscript. 

Response for reviewer 1

1. Most of the reasons for the incomplete immunization are same as in most of the studies. No doubt it confirms that the district in study of Ethiopia is also having similar reasons. As I understand poverty and illiteracy are the two main reasons in all the third world countries.

Response: Thank your dear reviewer for your comments. 

Response for reviewer 2

Major points

1. there are grammatical errors throughout the manuscript. I've tried to flag them but it may be smart to enlist the help of a text editor or native English speaker.

Response: Thank you dear reviewer for your comments, we have corrected the grammar, and spelling errors. 

2. What about missed opportunities of vaccination, i.e. children who did not get their vaccination at the exact right time for example due to a missed appointment, but a little while later?

Response: Thank you dear reviewer for clarification question. In this study, children who missed appointment and received their vaccine later are included in the study.

3. The authors state that "All eligible Cases and controls were identified by using health post EPI registration books of the selected rural kebeles and health center EPI registration books of the selected urban kebeles". This means children that are not EPI registration books cannot be part of the study as cases or controls. Can this have led to selection bias? What percentage of children born in the region are in the EPI books? If many are not, then the authors may be missing factor for incomplete immunization that are relevant for those children.

Response: Thank you dear reviewer for your clarification question. In Ethiopia all children started immunization registered on the EPI registration book. Children who are not started vaccine, not register on EPI registration book. Our study focus, children who vaccine completed and defaulter (incomplete). EPI registration is the ideal sample frame to get these children. is EPI Children who were not started vaccine are not our study population. In this study setting 88 % of children registered on the health centers and health posts EPI registration books. 12% of children not registered that means not started vaccination. They are not our study population 

4. This study focused mainly on factors relating to the mother. In earlier reports (from Nigeria), the husband’s role as well as their 'permission' to give vaccination was also important: https://onlinelibrary.wiley.com/doi/pdfdirect/10.1111/j.1440-1754.2010.01956.x. Why did the authors focus mainly on the mothers? Does caregiver, for example also imply fathers, or is the impact of fathers likely to be different in Ethiopia compared to Nigeria?

Response: Thank you dear reviewer for your clarification question. In this study, we included father as caregivers in “others” category because of small number. We stated “others” category definition under table 1, as footnote 

5. Why was the Dabat region chosen specifically for this study?

Response: Thank you dear reviewer for constructive comments. In Northwest Ethiopia incomplete vaccination, ranges from 7-24%. In this study setting, high proportion of children (23.1%) had incomplete vaccination. This is reason to conduct this study in this study setting. We incorporated this comments in the revised version of manuscript. Please see line 81-83. 

6. in tables 1-4 add p values for statistical tests checking if groups differed significantly on characteristics. For example caregiver age and wealth index seems to differ, would be good to know if these are significant differences. I think standard practice is to use t tests for continuous variables and chi-square tests for categorical ones.

Response: Thank you dear reviewer for invaluable comments and suggestions. We added p-value in all tables. 

7. at various points the age of child is mentioned. When was this measured? I assume it is the child's age at middlepoint of your study time (april/may 2021), so 1 may 2021?

Response: Thank you, dear reviewer, for your clarification question. The age of the children was measured at data collected date. The data collection period is April 1 to May 1 2021.

8. Many continous independent variables were dichotomized, what was the reasoning behind this and how are the results if this is not done?

Response: Thank again dear reviewer for your clarification question. In this study continuous variables are categorized based on pervious study categorization. Furthermore, we did analysis using continuous variables but we didn’t find any significant results. 

9. were there any missing data, or cases were respondents did not recall particular things? it is mentioned as a limitation but the extent to which it occured cannot be seen in the results.

Response: - Thank you dear reviewer for constructive comments. We included the comments in the limitation section of the study.

Minor points

- ll 26: revise grammar of sentence

- ll 29: change to "a community-based (...)"

- ll 32 "one dose routine" -> "one dose of the routine"

- abstract results: I prefer the notation (AOR: 2.3, 95% CI 1.3 to 5.5) for clarity but refer to journal guidelines to see what they prefer

- abstract results: perhaps present these in order of largest effect to smallest effect?

- ll 41 "children's whose place of delivery at home" -> "children who were delivered at home"

- ll 49: an->the

- ll 49: "proposed to provoke" -> "which provokes"

- ll 54: "cost effective" -> cost-effective

- ll 55 please provide a reference

- ll 58 children->child

- ll 59 please specify the abbreviations

- ll 61 "vaccine preventable" -> vaccine-preventable

- ll 61 "according to national" -> "according to the national"

- ll 61: Why does EPI reduce mortality and morbidity of the 'mother' from vaccinate preventable diseases? Later section the authors show TT (tetanus injection). Maybe the can explain that here

- ll 62 remove "as"

- ll 68-80 I would suggest to move a large part of this to the start of the introduction.

- ll 70: please provide a reference for "due in large portion to immunization"

- ll 81 please specify abbreviation

- ll 86 please specify that you are talking about Ethiopia

- ll 93 "reasons of not to" -> "reasons to not"

- ll 102 perhaps it is more insightful to mention total number of births in this time period?

- ll 100-102 can you explain the different between "kebels" and "kebeles" for those unaware (like myself) what they mean?

- ll 105: consider splitting this sentence up as it is very long and therefore hard to read. For example "E.g. All children aged between 12-23 months with mothers/ caretakers who had at least one dose of routine vaccination residing in Dabat district, were considered as a source population for both cases and controls. Those children aged between 12-23 months who did not complete the recommended vaccination before her/his 1ste birthday in selected kebels during the data collection were considered cases. Whereas those children aged between 12-23 months who completed the recommended vaccination before her/his 1ste birthday in selected kebeld during data collection period were considered as controls"

- ll 108 "her/his" -> "their"

- ll 110 "her/his" -> "their"

- ll 113 "from EPI" -> "from the EPI"

- ll 118 "for unmatched" -> "for an unmatched"

- ll 121 "a lot of" -> "many"

- ll 121 specify abbreviation

- ll 121 "follows up" -> "follow-ups"

- ll 122 remove "so,"

- ll 127 "by lottery" -> "by a lottery"

- ll 134 "Socio" -> "socio" (remove capital)

- ll 134: I take it these are independent variables as well? This is not immediately clear from the text.

- ll 138-139 please specify abbreviations

- ll 133-169 this may be converted to tables for clarity

- ll 149 "his/her" -> "their"

- ll 151: what about delayed vaccination i.e. missed opportunities?

- ll 171-172 remove "which were published in credential journals"

- ll 173 "in to" -> "into"

- ll 189 remove "were"

- ll 212 level->levels

- table 1: at occupation for controls, the 98.6% at gov't seems like an error as the percentages add up to more than 100%

- ll 283 remove "were"

- ll 240 Sorry I do not understand, round of what?

- ll 241 remove "to"

- table 3 please specify abbreviations.

- ll 248 "had good knowledge" -> "had a good knowledge"

- ll 253 I think this belongs in the caption of table 4?

- table 5 please specify abbreviations seperately in the caption

- table 5 first two columns (cases and controls) can be removed here as they already reported in earlier tables.

- ll 273 "but" -> "but a"

- ll 274 remove "was"

- ll 275 "increase them to die" -> "increases their risk of dying"

- ll 284 "also help by remembers" -> "can also help by remembering"

- ll 286 "was significant" -> "was a significant"

- ll 294 "was significant" -> "was a significant"

- ll 301 which->this

Response: Thank you, dear reviewer, for your constructive comments. We accepted all the minor comments and we have incorporated in the revised version of manuscript.

---

## [Decision Letter · Decision Letter 1]

30 Aug 2022

Determinants of incomplete childhood immunization among children aged 12-23 months in Dabat district, Northwest Ethiopia: Unmatched case- control Study

PONE-D-21-34551R1

Dear Dr. Boke,

We’re pleased to inform you that your manuscript has been judged scientifically suitable for publication and will be formally accepted for publication once it meets all outstanding technical requirements.

Kind regards,

Ricardo Q. Gurgel, PhD

Academic Editor

PLOS ONE

Additional Editor Comments (optional):

I have revised the response to reviewers and we had the decision for acceptance from one reviewer. The second reviewer could not revise the authors responses, but I have done it and I aggree to accept the manuscript for publication.

Reviewers' comments:

Reviewer's Responses to Questions

**Comments to the Author**

1. If the authors have adequately addressed your comments raised in a previous round of review and you feel that this manuscript is now acceptable for publication, you may indicate that here to bypass the “Comments to the Author” section, enter your conflict of interest statement in the “Confidential to Editor” section, and submit your "Accept" recommendation.

Reviewer #1: All comments have been addressed

2. Is the manuscript technically sound, and do the data support the conclusions?

Reviewer #1: Yes

3. Has the statistical analysis been performed appropriately and rigorously? 

Reviewer #1: Yes

4. Have the authors made all data underlying the findings in their manuscript fully available?

Reviewer #1: Yes

5. Is the manuscript presented in an intelligible fashion and written in standard English?

Reviewer #1: Yes

6. Review Comments to the Author

Reviewer #1: Thanks for correcting the manuscript. There are still few grammatical mistakes which needs to be corrected in final manuscript

7. PLOS authors have the option to publish the peer review history of their article (what does this mean?). If published, this will include your full peer review and any attached files.

Reviewer #1: No

---

## [Editor Report · Acceptance letter]

12 Oct 2022

PONE-D-21-34551R1 

Determinants of incomplete childhood immunization among children aged 12-23 months in Dabat district, Northwest Ethiopia: Unmatched case- control Study 

Dear Dr. Boke:

I'm pleased to inform you that your manuscript has been deemed suitable for publication in PLOS ONE. Congratulations! Your manuscript is now with our production department. 

Kind regards, 

on behalf of

Professor Ricardo Q. Gurgel 

Academic Editor

PLOS ONE